# N-Acetylcysteine to Reduce Kidney and Liver Injury Associated with Drug-Resistant Tuberculosis Treatment

**DOI:** 10.3390/pharmaceutics17040516

**Published:** 2025-04-15

**Authors:** Idu Meadows, Happiness Mvungi, Kassim Salim, Oscar Kaswaga, Peter Mbelele, Alphonce Liyoyo, Hadija Semvua, Athumani Ngoma, Scott K. Heysell, Stellah G. Mpagama

**Affiliations:** 1Division of Infectious Diseases and International Health, University of Virginia, Charlottesville, VA 22903, USA; pqd8ua@uvahealth.org (I.M.); skh8r@uvahealth.org (S.K.H.); 2Kibong’oto Infectious Disease Hospital, Siha 25102, Tanzania; happiness.mvungi@kidh.go.tz (H.M.); kassim.msaji@kidh.go.tz (K.S.); osca.kaswaga@kidh.go.tz (O.K.); peter.mbelele@kidh.go.tz (P.M.);; 3Kilimanjaro Christian Medical College, Moshi 25212, Tanzania; h.semvua@kcri.ac.tz

**Keywords:** rifampin-resistant pulmonary tuberculosis, standard rifampin-resistant treatment, N-acetylcysteine, adverse events, severe adverse events

## Abstract

**Background:** New drug classes and regimens have shortened the treatment duration for drug-resistant tuberculosis, but adverse events (AEs) and organ toxicity remain unacceptably common. N-acetylcysteine (NAC) has demonstrated potential in reducing kidney and liver toxicity in other clinical settings, but efficacy in drug-resistant tuberculosis treatment has not been rigorously evaluated. **Method:** A randomized controlled trial was conducted at Kibong’oto Infectious Diseases Hospital in Tanzania to assess the efficacy of NAC in reducing AEs in patients undergoing rifampin-resistant pulmonary tuberculosis treatment. Participants received an all-oral standardized rifampin-resistant regimen alone, with NAC 900 mg daily, or NAC 900 mg twice daily for 6 months. AEs, severe AEs, and renal and liver toxicity were monitored monthly and classified according to the Risk, Injury, Failure, Loss, and End-stage kidney disease criteria and National Cancer Institute Common Terminology Criteria for Adverse Events. Incident ratios and Kaplan–Meier curves were employed to compare group event occurrences. **Results:** A total of 66 patients (mean age 47 ± 12 years; 80% male) were randomized into three groups of 22. One hundred and fifty-eight AEs were recorded: 52 (33%) in the standard treatment group, 55 (35%) in the NAC 900 mg daily group, and 51 (32%) in the NAC 900 mg twice-daily group (*p* > 0.99). Severe AEs were observed in four patients in the standard group, two in the NAC 900 mg daily group, and three in the NAC 900 mg twice-daily group. Renal toxicity was more prevalent in the standard treatment group compared to those that received NAC (45% vs. 23%; *p* = 0.058), with a shorter onset of time to toxicity (*χ*^2^ = 3.199; *p* = 0.074). Liver injury events were rare across all groups. **Conclusion:** Among Tanzanian adults receiving rifampin-resistant tuberculosis treatment, NAC did not significantly reduce overall AEs but demonstrated important trends in reducing renal toxicity.

## 1. Introduction

Tuberculosis (TB) disease affects more than 10 million people annually [1]. Of these estimated in the 2023 Global TB Report, 410,000 people developed drug-resistant tuberculosis (DR)-TB leading to 160,000 DR-TB deaths. DR-TB is classified into two categories: rifampin-resistant or multi-drug-resistant (RR/MDR) and extensively drug-resistant (XDR) based on the approach to treating these resistance patterns. RR/MDR-TB is resistant to at least isoniazid and rifampin, the cornerstone medicines for the treatment of otherwise susceptible TB [2]. XDR-TB fulfills the definition of MDR/RR-TB with resistance to anti-TB fluoroquinolones and at least one additional group A drug (group A drugs are the most potent among those typically used for drug-resistant forms of TB using more extended treatment regimens and comprise levofloxacin, moxifloxacin, bedaquiline, and linezolid) [1,2,3,4]. Earlier, second-line DR-TB treatments included 20–24 months of drugs that were deemed less effective and more toxic [5,6]. Recent advancements, as seen in such trials as TB-PRACTECAL and ZeNix, have led to more efficacious and shorter-duration treatments. Currently, numerous trials and operational research studies are testing combinations of group A drugs and other recently introduced agents in varying durations for DR-TB [4].

While an intense focus has been placed on improving efficacy and reducing treatment duration, toxicity and intolerability remain a significant barrier to treatment success [7,8,9]. Adverse events (AEs) are unexpected negative responses to a medicine or vaccine exposure [7]. The proportion of people experiencing AEs during drug-susceptible TB treatment has varied widely (8–85%). However, AEs are more consistently reported among cohorts treated with DR-TB regimens (69–96%) [6]. In East Africa, AEs reported among cohorts in Uganda, Ethiopia, and Tanzania were 67.4% of 120 patients [6], 98.6% of 72 [6,10], with 87.7% [11]. of 260 patients analyzed, respectively. In DR-TB patients, AEs can result in interruptions, treatment failures, acquired resistance, prolonged hospital stays, or death [12], and while AEs in DR-TB treatment can affect any organ system they more commonly affect the kidney and liver [13,14,15].

N-acetylcysteine (NAC) has been trialed in different TB and non-TB clinical contexts for AE prevention due to its anti-oxidative and anti-inflammatory mechanisms of action. NAC acts as a thiol precursor to the amino acid L-cysteine, contributing to glutathione production and indirectly countering oxidative stress associated with glutathione depletion. Through the NAC-free thiol group, it also binds with active redox metal ions and can react with reactive oxygen and nitrogen species and can provide direct antioxidant activity [16,17,18]. NAC’s anti-inflammatory activity originates from inhibiting the pro-NF-kappa beta pathway, suppressing inflammatory cytokine release [17,18,19,20]. NAC is metabolized through the liver and excreted via the renal system. In healthy adults, the life span is 5.6 h [16,21].

Prior clinical uses of NAC include preventing contrast-induced renal toxicity in patients with myocardial infarction undergoing primary percutaneous coronary intervention [22,23], and liver injury prevention in drug-susceptible TB treatment, acetaminophen overdose, or alcohol-induced liver injury [12,16,17,18,19,24,25,26]. However, little is known about NAC’s protective effects against AEs in DR-TB treatment. To determine the potential clinical impact of NAC in reducing AEs associated with DR-TB treatment regimens, we conducted a randomized control trial among adults initiating pulmonary RR/MDR-TB treatment at Kibong’oto Infectious Disease Hospital, in Tanzania (PACTR202007736854169 registered on 3 July 2020). The primary objective was to assess the occurrence of clinical or laboratory-based AEs during the initial six months of therapy in patients utilizing a standardized all-oral, bedaquiline-based RR/MDR-TB regimen, the standard regimen and NAC of 900 mg oral daily, or the standard regimen and NAC 900 mg oral twice daily [12].

## 2. Materials and Methods

The inclusion criteria were: age range from 18–75 years; newly diagnosed RR or MDR-TB without additional fluoroquinolone or XDR-TB resistance pattern; no prior diagnosis or RR/MDR-TB treatment; a Karnofsky score of 50 or above defined as individuals requiring less considerable or frequent medical care; female participants had a negative urinary pregnancy test; kidney function with creatinine < 2 mg/dL creatinine or creatinine clearance level > 30 mL/min per the Cockcroft–Gault formula; hemoglobin > or 8 g/dL; platelet count > 100 k/mm^3^; AST, ALT, or bilirubin < 2× upper limit normal (ULN); and an electrocardiogram (EKG) with normal sinus rhythm at baseline. Exclusion criteria were: a known allergy or hypersensitivity to NAC; living with human immunodeficiency virus (HIV) that met the WHO clinical stage 4 disease criteria or had CD4 count < 100 cells/mm; history of cardiac arrhythmias or QTc prolongation; pregnant; exposed to IV fungal medication within the last 90 days; had previous or existing brain pathology (major head trauma, meningitis, encephalitis, metastasis, vestibular schwannoma); or had participated in other clinical trials with investigational agents within eight weeks prior to enrollment.

### 2.1. Ethical Considerations

Eligible participants were given a detailed review of the study protocol, and the study staff discussed and answered any concerns or questions. Participants provided written informed consent for the study, which was approved by the Tanzanian National Institute of Medical Research.

#### Consent for Publications

All authors provided consent to the publication and any data or images depicted within the manuscript and Appendix A.

### 2.2. Interventions

NAC 900 mg was procured in tablet form and manufactured by BioAdvantex Pharma Inc. (Mississauga, Canada). Randomization groups included the standardized RR/MDR-TB antibiotic regimen alone group, the standardized regimen plus NAC 900 mg orally once-daily group, and the standardized regimen plus NAC 900 mg orally twice-daily group. The standardized regimen, termed RISE-Removed Injectable Short-Course for eXpert-diagnosed RR/MDR-TB, was an all-oral regimen constructed from regional M. tuberculosis drug-susceptibility patterns and studied operationally at multiple sites in Tanzania [27]. Daily dosed bedaquiline was given for the first 6 months, linezolid for the first 2 months, and levofloxacin, clofazimine, pyrazinamide, and cycloserine for the entire duration of 9 months and extended up to 12 months if there was delayed clearance of *M. tuberculosis* from the sputum. Delaminid was used as an alternative if known resistance to one drug in the regimen or drug-specific intolerance developed. The Hain Genotype MTBDR-sl line probe assay was available for testing for common *gyrA* mutations conferring fluoroquinolone resistance.

Participants were actively followed weekly from treatment initiation through week 24 for symptom screening, targeted physical examination, and blood collection for clinical and laboratory AEs. Scheduled complete blood counts and comprehensive metabolic panels were performed at treatment initiation (baseline), at week 2, at week 4, and every four weeks thereafter through week 24. EKGs were obtained at baseline, week 1, week 2, week 4, and then every four weeks through week 24. The complete blood count was used to determine anemia, while a comprehensive panel was used to determine liver and kidney toxicity. A chest X-ray was obtained at baseline and week 24. Per routine clinical practice, sputum was collected at baseline and weekly until the acid-fast bacilli (AFB) smear and mycobacterial culture were negative.

AE severity was graded according to the Common Terminology Criteria for AEs (CTCAE) version 5.0 from the U.S. Department of Health and Human Services National Cancer Institute, published in November 2017. Renal toxicity was graded with Risk, Injury, Failure, Loss, and End-stage kidney disease (RIFLE) for early identification of renal toxicity. Detailed definitions and criteria for adverse events are available in Appendix A.

### 2.3. Outcome Measures

The primary outcome measure was the total number of AEs and severe AEs (SAEs) in each group through 26 weeks which was two weeks after the final study medication doses. The secondary outcomes, including the total number and time to events of renal toxicity, liver toxicity, and anemia, were chosen based on the expected higher frequency and prior evidence from other settings that NAC may prevent or delay drug-induced renal and liver toxicity. While there was no expected impact on anemia, this planned measurement served as an internal control check against a spurious finding.

### 2.4. Statistical Analysis

The primary outcome was comparing the incidence of total AEs and SAEs in each arm, which was calculated using counts and proportions. Differences in the incidence of total AEs and SAEs and individual categories of AEs among each group were analyzed using the Pearson Chi-square and Fisher Exact tests. For secondary renal and liver injury and anemia outcomes, both NAC treatment groups were combined and compared to the standard treatment group with Kaplan–Meier estimates plotted for time-to-event analysis. The statistical analyses were performed using IBM SPSS Statistics for Windows, Version 29. *p* values of <0.05 were considered statistically significant.

#### Metadata

Metadata are available upon request.

## 3. Results

From 2020 to October 2022, 70 patients were screened, and 66 met the eligibility criteria (Figure 1). Sixty-six were randomized to standardized RR/MDR-TB treatment (*N* = 22), standardized treatment with NAC 900 mg daily (*N* = 22), and standardized treatment with NAC 900 mg twice daily (*N* = 22) (Figure 1). The average age was 47 years with a standard deviation (SD) ± 12 years. There were no differences in baseline creatinine, AST, ALT, and hemoglobin among the randomization groups (Table 1). Seven patients (10.6%) withdrew from the study, six voluntarily and one due to pregnancy. Two additional patients died during the study: one patient in the NAC daily group and one patient in the NAC twice-daily group.

### 3.1. Incidence of Adverse Events

There were 158 total adverse events (AEs), including 52 (33%) in the standard treatment group, 55 (35%) in the NAC 900 mg daily group, and 51 (32%) in the NAC 900 mg twice-daily group, *p* > 0.99 (Table 2). Events of renal toxicity occurred in 10 participants (45%) in the standard treatment group, six (27%) in the daily NAC group, and four (18%) with twice-daily NAC (Table 2). Assessing any participant that received NAC, there were fewer total events of renal toxicity (10 events, 23% of all 44 participants that received NAC) compared to the standard of care group (10 participants with events, 45%), *p* = 0.058 (Appendix A). AEs of liver toxicity occurred less commonly in 9% (two of 22) in each group. Events of anemia were similarly distributed across groups, occurring in seven (32%) in the standardized RR/MDR-TB regimen group without NAC, seven (32%) in the daily NAC group, and nine (40%) in the twice-daily NAC group.

Total SAEs occurred in six participants (27%) in the standardized RR/MDR-TB regimen group without NAC, and four (18%) in both the NAC once-daily and NAC twice-daily groups. SAEs related to renal toxicity occurred in four participants (18%) in the standard treatment group that did not receive NAC and in only one participant each (5%) in the once-daily and twice-daily NAC treatment groups. There were no SAEs of liver injury in any of the groups. SAEs for anemia occurred in one (5%) of the participants in the standardized RR/MDR-TB regimen without NAC, one (5%) in the daily NAC group, and two (9%) in the twice-daily NAC group.

### 3.2. Time to Individual Adverse Events

A log-rank test determined if there were differences in the time to event of the pre-specified secondary outcomes of interest (AEs for renal toxicity, liver toxicity, or anemia) among those that received NAC (combined NAC daily and NAC twice-daily groups) compared to those that did not. The time to renal toxicity was shorter and AEs of renal toxicity continued to occur later in the treatment course for those not receiving NAC, *χ*^2^(1) = 3.199, *p* = 0.074 (Figure 2). There was no difference in time to events for liver toxicity or anemia (Appendix A).

## 4. Discussions

In this phase 2b randomized controlled trial among adults starting RR/MDR-TB treatment once or twice daily, NAC did not significantly reduce the total number of AEs or SAEs over the first 26 weeks of treatment compared to placebo. However, important trends suggest that NAC may reduce the number, severity, and time to events of renal toxicity. There was no difference in the total number or time for the development of liver injury among the treatment groups. Nevertheless, the overall incidence of liver injury in the study population was lower than expected.

NAC has been found to protect against acute renal toxicity in high-risk populations with diabetes, cardiac disease, chronic kidney disease, or other chronic illnesses during events such as contrast exposure, coronary artery bypass surgery, and kidney transplantation [22,23,28,29,30,31]. Most explanations of NAC’s effects on renal toxicity suggest that it prevents renal vasoconstriction, inflammation, and oxidative stress. NAC increases endothelial nitric oxide synthase expression, nitric oxide, and prostaglandin E2 production and reduces angiotensin-converting enzyme activities, thus preventing renal vasoconstriction [22,32,33]. NAC has also been demonstrated to reduce oxidative stress related to ischemic reperfusion by eliminating free radicals and indirectly generating glutathione. Additionally, NAC may down-regulate inflammation by inhibiting the transcription of activator protein one and nuclear factor kappa-light-chain-enhancer and reducing lymphocyte and macrophage infiltrations [21,23,28,29,30,31,32,34]. Thus, the putative activity of NAC is biologically plausible to explain the observed decrease in events of renal toxicity that could accumulate with multiple drugs over a long period in RR/MDR-TB treatment and explain both the prevention of early- and more late-onset kidney toxicity.

Interestingly, the overall incidence of renal toxicity in 30% of the total study population was higher than other trials in RR/MDR-TB. For instance, the ZeNix trial reported 13.3% renal toxicity in 45 participants who received bedaquiline, pretomanid, and linezolid for 26 weeks [9]. In the TB PRACTECAL trial, only 2.8% of 142 participants experienced renal toxicity during treatment with a 24-week regimen of bedaquiline, pretomanid, linezolid, and moxifloxacin [35]. The variation in renal toxicity incidence may be explained by factors such as baseline or reference serum creatinine, the inclusion/exclusion criteria for these trials of new drug regimens, and the difference in renal toxicity diagnostic criteria used in each study [36]. For instance, these studies used the Division of Microbiology and Infectious Disease Table to grade renal toxicity severity; in contrast, we used the RIFLE criteria, a more pragmatic approach for early identification of participants at risk of developing renal toxicity risk, which likely led to the classification of more patients with renal toxicity in our study [36,37]. Another explanation was the difference in the median age of our study which was older at 47 compared to both studies, ZeNix and TB-PRACTECAL trials, where the median ages were 34 and 38, respectively, with increasing age related to decreased drug clearance rate and risk of renal toxicity [10]. Another explanation is the difference in drug composition of regimens used, where our study included pyrazinamide and cycloserine, which both have been reported to cause higher rates of renal toxicity [38].

Prior to this current trial in RR/MDR-TB, NAC has been used to prevent against liver toxicity in drug-susceptible TB treatment, where first-line drugs with significant liver injury profiles are used in combination (isoniazid, rifampin, and pyrazinamide). The hepatoprotective effect is mainly related to NAC’s promoting glutathione regeneration, detoxifying harmful reactive oxygen species, and reducing oxidative stress [24,25,26,39,40,41,42]. In a study by Baniasadi et al. in 2010 [40], 60 patients with an average age of 60 or older and newly diagnosed TB were randomly assigned to two groups. Twenty-eight patients received oral NAC 600 mg twice daily in addition to drug-susceptible TB treatment, while 32 patients in the control group received only drug-susceptible TB antibiotics [40]. None of the patients in the intervention group developed liver toxicity, whereas 17 (53%) patients in the control group developed liver toxicity within two weeks of evaluation. Similarly, in a study by Ahmed et al. in 2020, 162 patients aged 18 to 60 with TB were divided into two groups per treatment arm [24]. Half of the patients (81 of 162) in the control group received DS-TB treatment without NAC, while the other half in the intervention group received both NAC 600 mg twice daily with DS-TB for eight weeks. In the control group, 17 (21%) patients developed liver injury two weeks after the initiation of DS-TB treatment, compared to only one patient (1%) in the interventional group. As a result, each study concluded by recommending NAC in combination with drug-susceptible TB treatment to reduce liver toxicity, a practice that has not been widely adopted. We found less liver injury and so could not distinguish a benefit of NAC, and were likely underpowered to do so. We suspect there are fewer events of liver toxicity given that rifampin and isoniazid were not part of this study regimen which may accelerate liver injury that is not observed in most RR/MDR-TB regimens [10,43,44]. Our study population also had fewer women than in other trials of new RR/MDR-TB regimens. It has been suggested that people of female sex are more prone to liver injury compared to males in other studies of TB treatment due to body composition, enzyme activity, and hormonal regulation [44]. Similarly, other studies of more undernourished populations compared to our current study also suggest additional risk of TB treatment-induced liver injury [45]. To gain more conclusive results, a larger trial involving a more balanced gender distribution to determine whether sex or other biomarkers influence the efficacy and safety of RR/MDR-TB regimens may be necessary to observe any potential protective effects on the liver from NAC.

While we tested two dosing regimens of NAC, the total bioavailability may have been too low to produce the maximum impact [46]. Usual oral dosing of NAC have ranged from 600–1200 mg and up to three times daily, but even higher dosages have been used in psychiatric illnesses. Intravenous (IV) NAC reaches a maximum concentration of 500 mg/L within 15 min of 150 mg/kg administration, and IV dosing ranges from 50 mg/kg to 150 mg/kg as used in acetaminophen overdose, neurodegenerative diseases, post-cardiac surgery, and contrast-induced acute kidney injury [16]. In our study, we used effervescent oral NAC at doses 900 mg daily and 1800 mg split over twice-daily administration and these dosages are within frequently used ranges for anti-inflammatory and anti-oxidative effects. In Green et al., effervescent NAC demonstrated bioequivalence to oral NAC, and we chose effervescent because of its feasibility compared to IV NAC and tolerability compared to the oral pill form of NAC [47]. Nevertheless, this is the first trial to our knowledge of long-term (24-week) NAC administration, and our findings suggest that the intracellular benefits of NAC are important for offering protection over the entirety of the treatment duration as toxicity events continued throughout all weeks of antibiotic administration. Future studies could consider a higher dose of oral or IV formulations administered in the first week of therapy that could be transitioned to a more conventional oral dosing regimen concurrent with the remainder of TB treatment, or varying durations of NAC to determine if protection continues beyond NAC discontinuation.

Following the TB treatment interval, NAC carries a potential as an approach to preventing post-TB lung disability (PTLD)—a debilitating respiratory condition estimated to impact half of TB survivors after cure [48]. One postulated pathophysiology of PTLD is an imbalance in inflammatory and oxidative responses following TB disease and treatment [48,49,50]. Given NAC’s role in immune modulation and replenishing glutathione, breaking disulfide bonds, and restoring thiol stores, NAC may prevent the adverse remodeling and fibrotic response to tissue injury during and following TB treatment [17]. A recent study by Wallis et al. (2024) investigated the effects of NAC as an adjunctive therapy in patients with drug-susceptible TB that had developed PTLD and found that NAC treatment led to improvements in lung function [51]. This is not entirely unexpected as NAC is currently used to improve persistent chronic obstructive pulmonary disease exacerbations [52] and has demonstrated efficacy in increasing survival in the primary composite end point of death, hospitalization, or decline in lung function for individuals with idiopathic pulmonary fibrosis associated with the rs3750920 (*TOLLIP*) TT genotype [53]. As such, we plan for further follow-up of randomized groups in this current trial to determine the long-term benefits of NAC including prevention of PTLD.

Several limitations affected the generalizability and interpretation of the trial. The study was not designed to evaluate optimal NAC dosages or serum concentrations on circulating glutathione levels. While two different dosing regimens were tested, the primary outcome was clinical and laboratory-measured AEs, and the relationship between specific AE development and glutathione level was unknown. Furthermore, while consistent trends in the prevention of AEs and SAEs from renal toxicity and the time to renal toxicity were observed in patients receiving NAC, secondary assessments were not performed of cystatin-C or other non-enzymatic measurements of kidney function to determine if enzymatic interference could have occurred with high NAC exposures [54,55]. As discussed, sex-related differences in renal clearance and liver enzyme activity may influence drug metabolism and the occurrence of adverse events. Given that our study population was approximately 80% male, it may not have fully captured these potential sex-based variations in drug metabolism and toxicity.

In conclusion, this is the first randomized study to evaluate the effects of NAC in reducing adverse events associated with RR/MDR-TB treatment. Our findings suggest that NAC may have potential in reducing renal toxicity—an adverse event that often leads to hospitalization, treatment interruption, or regimen modification. To confirm these findings, a larger-scale clinical trial is warranted and could be powered on the current trial’s effect size in reducing events of renal toxicity. Future studies should also incorporate a dose-finding lead-in to evaluate NAC plasma exposure and glutathione levels. Additionally, trial outcomes should include non-enzymatic kidney function assessments alongside conventional creatinine measurements, with a gender-balanced study design and extended follow-up to determine NAC’s long-term impact on lung health post-TB cure.

## Figures and Tables

**Figure 1 pharmaceutics-17-00516-f001:**
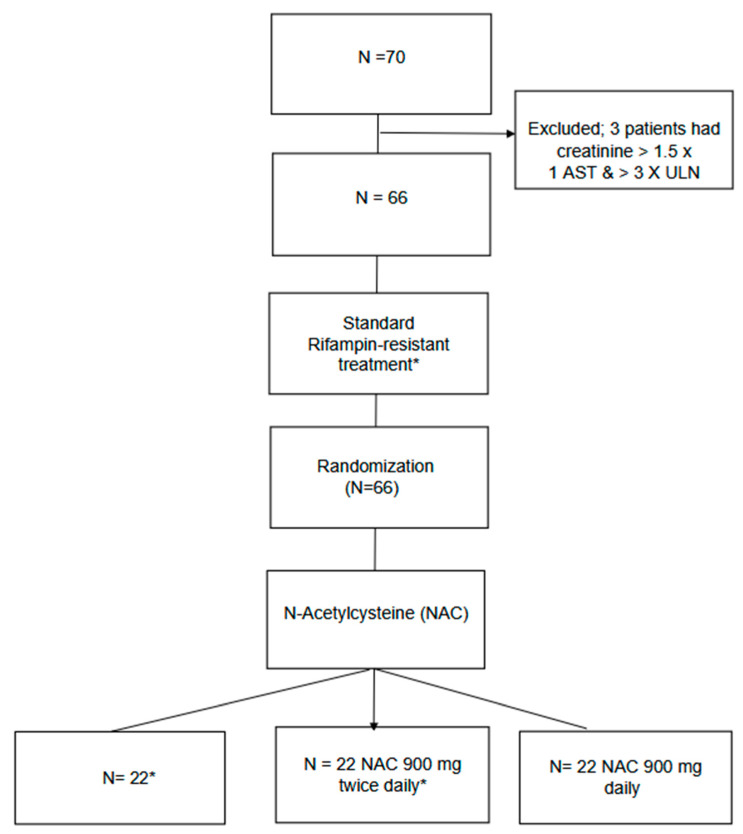
Study flow chart displaying enrollment and randomization of study participants. * Bedaquiline, Levofloxacin, cysloserine, clofazimine, pyrazinamide, Linezolid, alternative: Delaminid. aspartate aminotransferase (AST), Alanine aminotransferase (ALT), upper limit normal (ULN).

**Figure 2 pharmaceutics-17-00516-f002:**
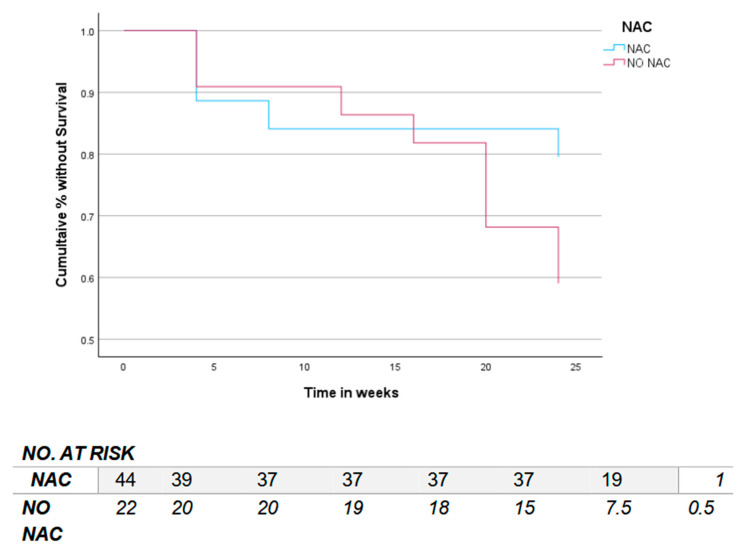
Time to event curve of renal injury through 24 weeks of standard treatment and NAC group. Note: N-acetylcysteine (NAC). Number at risk of renal injury without an event (NO. AT RISK) in those receiving NAC and those without (NO NAC) at each time of measurement (0 weeks-baseline, 4, 8, 12, 16, 20 and 24 weeks after enrollment).

**Table 1 pharmaceutics-17-00516-t001:** Demographics and baseline laboratory results of study participants (*N* = 66).

Treatment Arm (*N* = 66)	Control (*N* = 22)	Daily (*N* = 22)	Twice-Daily (*N* = 22)
Age ± SD	47.8 ± 12.4	47.5 ± 12.3	47.7 ± 12.3
BMI ± SD	19.11 ± 2.7	19.05 ± 2.7	19.12 ± 2.69
Gender male (%)	17 (77)	18 (81)	18 (81)
Smoking (%)	9 (41)	9 (41)	9 (41)
HIV (%)	5 (23)	2 (9)	4 (18)
AST median (IQR)	27.8 (22.1–40.1)	28.4 (21.3–34.8)	34.8 (24.0–69.0)
ALT median (IQR)	21.8 (10.1–28.7)	20.3 (13.8–27.4)	26.6 (22.1–30.8)
Creatinine median (IQR)	76.4 (63.0–90.0)	79.9 (62.8–94.4)	82.8 (69.1–106.0)
Hemoglobin median (IQR)	13.5 (12.2–15.1)	14 (12.7–14.9)	13.0 (11.18–15.0)

Note: N-acetylcysteine (NAC) given 900 mg once daily or twice daily. Alanine aminotransferase (ALT), aspartate aminotransferase (AST), human immunodeficiency virus (HIV), body mass index (BMI), interquartile range (IQR), standard deviation (SD), number of participants (N).

**Table 2 pharmaceutics-17-00516-t002:** Incidence of AEs in interventional group vs. standard treatment group of (*N* = 66) participants.

System	TAEs in Standard Treatment Group	Total Patients with at Least One Event in Standard Treatment Group (*N* = 22)	TAEs in NAC Daily Group	Total Patients with at Least One Event in Daily (*N* = 22)	TAEs Twice-Daily (AEs)	Total Patients with at Least one Event in Twice-Daily (*N* = 22)	TAEs in All Patients Across all Groups	Total Patients with at Least One AE Across All Groups (*N* = 66)	*p*-Value	*p*-Value
central nervous	0	0	1	1 (5%)	1	1 (5%)	2	2	1.000	1.000
visual	0	0	2	2 (9%)	0	0	2	2	0.323	0.323
endocrine	2	2 (9) %	1	1 (5%)	0	0	3	3	0.767	0.767
gastro intestinal tract	5	3 (14%)	4	4 (18%)	11	3 (14%)	20	10	0.578	0.578
hepatic	2	2 (9%)	2	2 (9%)	2	2 (9%)	6	6	1.000	1.000
renal	16	10 (45%)	11	6 (27%)	6	4 (18%)	33	20	0.134	0.134
muscular-skeletal	15	6 (27%)	24	6 (27%)	19	7 (32%)	58	19	0.703	0.703
skin	3	3 (14%)	1	1 (5%)	0	0	4	3	0.312	0.312
hematology	9	7 (32%)	9	7 (32%)	12	9 (40%)	30	23	0.715	0.715
total AEs	52 (32%)		55 (35%)		51 (32%)		158		1.000	1.000
SAEs grade 3 or above										
renal	4	3 (14%)	3	1 (5%)	1	1 (5%)	8		0.606	0.606
hepatic	0	0	0	0	0	0	0			
hematology	2	1 (5%)	1	1 (5%)	3	2 (9%)	4		1.000	1.000
Total SAEs	6		4		4				0.901	0.901

Note: Comparing the incidence of total adverse events (TAEs) and severe AEs (SAEs) between the standard treatment group, daily, and twice-daily NAC using Fisher exact/*χ*^2^. N-acetylcysteine (NAC).

## Data Availability

Metadata are available upon request.

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
