# Peer review of "N-Acetylcysteine to Reduce Kidney and Liver Injury Associated with Drug-Resistant Tuberculosis Treatment"

_pharmaceutics, 2025, doi:10.3390/pharmaceutics17040516_

Round 1
Reviewer 1 Report
Comments and Suggestions for Authors
This manuscript focuses on the use of N-acetylcysteine (NAC) to prevent adverse events (AEs) in the treatment of drug-resistant tuberculosis (DR-TB). The study presents valuable insights into a novel approach for mitigating renal toxicity in patients undergoing RR/MDR-TB treatment, a major concern in TB management. Several areas need to be addressed to improve clarity and strengthen the comprehensiveness of this study.
Comments:
- Please specify the AE and SAE criteria/indication in the methods section.
- Liver and Anemia Toxicity: Is the low incidence of liver toxicity in this population a result of the drug regimen itself, or do the baseline characteristics of the cohort contribute to the low event rate?
- Although the author mentioned the use of NAC in other contexts (e.g., drug-induced liver injury and contrast-induced renal toxicity), a detailed review of previous studies in DR-TB or other similar settings would benefit this manuscript.
- This is a small sample size study and lacks of specific biomarker analysis, these limitations should be more clearly addressed and further discussed in the discussion. Besides, many factors are not being controlled that may influence the results interpretation, a multivariate analysis on the effect of NAC would benefit this manuscript.
- The statement "NAC appeared to prevent more late-onset kidney toxicity". If there is a trend toward prevention, please discuss how these trends can inform future studies, especially in larger cohorts.
Author Response
Reviewer #1:
This manuscript focuses on the use of N-acetylcysteine (NAC) to prevent adverse events (AEs) in the treatment of drug-resistant tuberculosis (DR-TB). The study presents valuable insights into a novel approach for mitigating renal toxicity in patients undergoing RR/MDR-TB treatment, a major concern in TB management.
Reply: Thank you for recognizing the valuable insights and novel approach to a significant global problem.
Several areas need to be addressed to improve clarity and strengthen the comprehensiveness of this study.
Please specify the AE and SAE criteria/indication in the methods section:
Reply: We agree and the criteria are discussed in lines 201 to 205 of the manuscript (Revision 1 Clean version). Additionally, we now cite Supplemental Table 1, which provides more detailed information.
Liver and Anemia Toxicity: Is the low incidence of liver toxicity in this population a result of the drug regimen itself, or do the baseline characteristics of the cohort contribute to the low event rate?
Although the author mentioned the use of NAC in other contexts (e.g., drug-induced liver injury and contrast-induced renal toxicity), a detailed review of previous studies in DR-TB or other similar settings would benefit this manuscript.
Reply: The low incidence of liver toxicity is largely attributed to our drug regimen, which included drugs with potential hepatoxicity such as bedaquiline, pyrazinamide and the flouroquinolone class, but predictably less hepatotoxic than drug-susceptible TB regimens that include isoniazid and rifampin. Please refer to the discussion section (lines 316–346) of the manuscript, where we further discuss the effects of NAC in preventing liver toxicity in the drug-susceptible TB population. In this extensive discussion, we also highlight that undernutrition and female sex have predisposed to hepatoxicity in other TB studies but were proportionally less common in our trial and may also explain the lower overall rate of hepatoxicity across groups. Nevertheless, we highlight that our study is the first randomized trial of different doses of NAC among people with MDR-TB, and further studies could be performed among the growing number of novel regimens used for MDR-TB.
-This is a small sample size study and lacks specific biomarker analysis. These limitations should be more clearly addressed and further discussed in the discussion. Besides, many factors that may influence the results' interpretation are not controlled. A multivariate analysis of the effect of NAC would benefit this manuscript.
Reply: We agree that there could have been further biomarker analysis (such as with glutathione levels) incorporated into this small randomized controlled trial that was powered only for detecting a difference in the total number of adverse events, with planned secondary analysis of common organ toxicities (e.g. nephrotoxicity). Importantly, the biologically plausible trend observed in prevention of nephrotoxicity incidence and nephrotoxicity severity now allows the powering of a larger randomized trial for this main which could include multivariate or other analytical methods to better understand predictors of the main effect. We further detail this potential in lines 343-346 and 402-408 of the manuscript.
-The statement "NAC appeared to prevent more late-onset kidney toxicity." If there is a trend toward prevention, please discuss how these trends can inform future studies, especially in larger cohorts.
Reply: The statement that 'NAC appeared to prevent more late-onset kidney toxicity' is based on the observation that the groups that received NAC had fewer cases of nephrotoxicity later in the treatment course but participants that did not receive NAC continued to experience nephrotoxicity events from week 19 onward (see Kaplan-Meier curve). Given this important trend, larger studies could explore not only the primary endpoints (kidney/liver toxicity and anemia) but also assess the time-to-event of these outcomes among groups with different NAC dosages but also varied durations (e.g. NAC given for the first 8 weeks of therapy compared to the entire treatment course). See revised lines 363-366 in the Discussion.
Reviewer 2 Report
Comments and Suggestions for Authors
Dear Authors,
After thorough evaluation, I acknowledge the merit of your manuscript; however, it does not fully meet the publication criteria for Pharmaceutics in its current form. Several shortcomings must be addressed before it can be considered for publication. Therefore, we invite you to submit a revised version that incorporates the necessary revisions based on the review feedback.
Your manuscript, titled "N-acetylcysteine to Reduce Kidney and Liver Injury Associated with Drug-Resistant Tuberculosis Treatment," presents the first randomized study investigating NAC's potential in mitigating adverse effects related to RR/MDR-TB treatment. The study suggests that NAC may help reduce renal toxicity—an issue that frequently results in hospitalization, treatment discontinuation, or regimen modifications.
The topic is highly relevant, and the experimental design is well-structured. However, significant revisions are required before the manuscript can be considered for publication.
Specific Comments:
- While the introduction effectively summarizes the study’s significance, incorporating a brief mention of potential clinical applications would enhance its impact.
- Consider adding tables summarizing existing research on the anti-TB effects of NAC and relevant clinical trials to strengthen the rationale for your hypothesis.
- Emphasize any novel insights or unique contributions that distinguish this study from existing research.
- Clearly outline the criteria used to assess successful model establishment. Additionally, include a figure illustrating the potential mechanisms through which NAC mitigates adverse effects associated with RR/MDR-TB treatment.
- The literature review is insufficient and outdated. Ensure that your references are current and relevant, particularly incorporating recent advancements that contextualize your findings within the broader research landscape. Avoid repetitive or overlapping content between the introduction and discussion sections.
- Increase the sample size (ideally to around 250 participants) to provide a more robust justification for the proposed model’s applicability.
- Since this study is conducted among Tanzanian residents, it is unclear whether the nomogram model applies to other ethnic groups. Future research should assess its suitability in more diverse populations.
- There are multiple grammatical errors and awkward phrasings throughout the manuscript. A thorough language revision, possibly with the assistance of a professional editor, is strongly recommended.
We look forward to receiving a revised version that addresses these concerns.
Comments on the Quality of English Language
There are several grammatical errors and awkward phrasings throughout the manuscript. I suggest a thorough revision of the manuscript for language issues, perhaps with the help of a professional editor.
Author Response
Reviewer #2
Your manuscript, titled "N-acetylcysteine to Reduce Kidney and Liver Injury Associated with Drug-Resistant Tuberculosis Treatment," presents the first randomized study investigating NAC's potential in mitigating adverse effects related to RR/MDR-TB treatment. The study suggests that NAC may help reduce renal toxicity—an issue that frequently results in hospitalization, treatment discontinuation, or regimen modifications. The topic is highly relevant, and the experimental design is well-structured.
Reply: Thank you for appreciating the considerable relevance and importance to test the intervention in a randomized trial design, indeed the first of its kind for MDR-TB toxicity prevention.
However, significant revisions are required before the manuscript can be considered for publication.
Specific Comments:
-While the introduction effectively summarizes the study’s significance, incorporating a brief mention of potential clinical applications would enhance its impact.
Reply: Thank you for your candid review. We have revised the final paragraph of the Introduction to clearly state the current uses of NAC and how this first-of-kind study in RR/MDR-TB would have potential to reduce similar types of adverse events that have been studied with NAC in other clinical contexts (nephrotoxicity and hepatoxicity) which separate from causing treatment interruption and delay in treatment of RR/MDR-TB can carry high morbidity and mortality. See new lines 134-150.
-Consider adding tables summarizing existing research on the anti-TB effects of NAC and relevant clinical trials to strengthen the rationale for your hypothesis.
Reply: We did not include a table summarizing existing research on the anti-tuberculosis effects of NAC, as such tables are more commonly included in a systematic review and not a randomized control trial design. However, we do specifically mention briefly in the revised Introduction the studies of NAC for hepatoxicity prevention in drug-susceptible TB. However, we agree with the reviewer and despite the fact there have been no previous trials of NAC in RR/MDR-TB, we explain in detail the previous findings of studies with drug-susceptible TB regimens in the fourth paragraph of the Discussion (see lines 321-351) and compare our trial with that existing literature, such as critical recent studies from Ahmed et al. (2020), Sanabria-Cabrera et al. (2022), and Sukumaran et al. (2023).
Furthermore, while data on NAC’s effects on renal function in TB are limited, we have discussed the general mechanisms by which NAC reduces renal toxicity in other non-TB populations in the second paragraph of the Discussion section (lines 281–295). Specifically, studies such as Huang JW et al. (2021), Guo Z et al. (2020), Xie W et al. (2021), Javaherforooshzadeh F et al. (2021), Modarresi A. et al. (2017), and Cepaityte et al. (2023).
-Emphasize any novel insights or unique contributions that distinguish this study from existing research.
Clearly outline the criteria used to assess successful model establishment. Additionally, include a figure illustrating the potential mechanisms through which NAC mitigates adverse effects associated with RR/MDR-TB treatment.
Reply: We agree with the proposed emphasis and have further revised the Introduction to place this research in context, as well as new revisions in the Discussion to describe the further studies that should continue from this trial’s findings. We also provide extensive detail on the mechanisms of NAC nephrotoxicity prevention (Discussion paragraph two) and the dose selection of NAC for the trial based on prior dose and delivery mechanism studies (Discussion paragraph four) that informed our model of study. Despite the unconventionality of including a mechanistic figure in a randomized control trial manuscript as would be more common in a review article, we did consider this but ultimately did not include given that we did not measure new biomarkers in the NAC pathway of toxicity prevention (e.g. glutathione, see Discussion paragraphs 6 and 8) and deferred to convention of only presenting new data generating in this trial for tabular or figure presentation. To summarize for the Reviewer to direct specifically to lines in the manuscript:
- Renal: NAC prevents renal vasoconstriction, inflammation, and oxidative stress by increasing endothelial nitric oxide synthase expression, nitric oxide, and prostaglandin E2 production while reducing angiotensin-converting enzyme activities. It also mitigates oxidative stress from ischemia-reperfusion and drugs by eliminating free radicals and promoting glutathione production. Additionally, it reduces inflammation by inhibiting transcription factors (activator protein-1 and nuclear factor kappa-light-chain-enhancer) and limiting immune cell infiltration. Please refer to line (281-295)
- Liver: NAC counteracts oxidative stress through both direct and indirect antioxidant mechanisms. Indirectly, it serves as a precursor to L-cysteine, aiding glutathione production to counteract oxidative stress. Directly, its free thiol group binds with redox metal ions and neutralizes reactive oxygen and nitrogen species. Please refer to line 316 to 346.
- Pulmonary (especially in tuberculosis): NAC may prevent adverse lung remodeling and fibrosis following tuberculosis treatment due to its role in immune modulation, glutathione replenishment, disulfide bond breaking, thiol store restoration, and enhancement of anti-TB drug effects. It is already used to manage persistent chronic obstructive pulmonary disease exacerbations and cystic fibrosis. Please refer to line 368-383.
The literature review is insufficient and outdated. Ensure that your references are current and relevant, particularly incorporating recent advancements that contextualize your findings within the broader research landscape. Avoid repetitive or overlapping content between the introduction and discussion sections.
Reply: We have revised the Introduction and Discussion sections to minimize overlapping content. The majority of our references date from 2020 onwards (please see response to prior Reviewer queries), and we now also have conducted a further literature review to strengthen the discussion on NAC’s potential protective effects on relevant organ systems. This includes the addition of recent studies, such as Wallis et al. 2024 for NAC effect on pulmonary toxicity during and after drug-susceptible treatment, and Hernandez-Cruz et al 2024, Haami et al 2023 and Popescu et al 2024 for related nephrotoxicity and hepatoxicity benefits.
Increase the sample size (ideally to around 250 participants) to provide a more robust justification for the proposed model’s applicability.
Reply: We appreciate the reviewer’s insight, but the randomized controlled trial is closed to enrolled and completed in analysis, it is therefore not feasible to increase the number of study participants. However, larger studies could consider expanding the sample size to power for the reasonable effect size our trial has found for prevention of nephrotoxicity events as the primary outcome. We had revised the Discussion to emphasize this point.
-Since this study is conducted among Tanzanian residents, it is unclear whether the nomogram model applies to other ethnic groups. Future research should assess its suitability in more diverse populations.
Reply: NAC has been tested in various populations despite the heterogeneity associated with disease progression. In the previous studies we reviewed, including those from sub-Saharan Africa—such as Ejigu et al. (2020), Tenorio et al. (2021), Mpamba et al. (2022), and more recently, Wallis et al. (2024) on NAC’s effects on the lungs, as well as Hernandez-Cruz et al. (2024), Haami et al. (2023), and Popescu et al. (2024)—NAC’s effects did not appear to be significantly associated with ethnicity. NAC treatment is widely used in the North America and Europe to prevent liver toxicity associated with acetominophen overdose. Additionally, it serves as an adjunctive therapy in cystic fibrosis—primarily affecting individuals of European descent—by breaking down excessive mucus production (see Guerini et al., 2022, and Calverley et al., 2021). While responses to different disease processes may vary across ethnic groups, NAC itself has not been shown to have ethnicity-dependent effects. Since our study was the first of its kind in the RR/MDR-TB population, future studies could be designed for diverse populations with RR/MDR-TB in other TB endemic regions outside of Tanzania to further validate the generalizability of the nomogram model.
There are multiple grammatical errors and awkward phrasings throughout the manuscript. A thorough language revision, possibly with the assistance of a professional editor, is strongly recommended.
Reply: The manuscript was written by English speakers in Tanzania and the United States. In this revision, we have extensively corrected minor grammatical errors and phrasing conventions to adhere to a Northern European phraseology. One author, as a prior English language and literature major, has further refined the style, and additionally carries experience in mentoring new scientific writers globally through federally sponsored training grants for enhancing scientific communication.
Reviewer 3 Report
Comments and Suggestions for Authors
This work by Meadows and colleagues adresses the use of NAC to reduce kidney and liver injury in MDR-TB patients, presenting a randomized trial investigating whether NAC mitigates averse events or not.
The authors provide a well-structured background, clearly defining the rationale for NAC use. The methodology is generally well-detailed, and the study is adequately powered for exploratory analysis. Some issues need to be clarified at this stage, please see below.
Major comments
- How were AE identified? Specifically, how did the atuhors identify situations of "renal toxicity, liver toxicity, anemia"? Through blood works? Other? This needs to explicited, I could only find information regarding the classification of the severity of the AE.
- No statistically signficant difference was found between the various groups, which in itself is OK, but the authors highlight a “trend” toward renal toxicity reduction with NAC (23% vs. 45%, p = 0.058). The confidence interval for this p value likley includes the possibility of both no effect and harm. The authors should revisist the statistic of this study, presenting for eample confidence intervals, without which it is difficult to assess the precision of estimates.
- Also, the study population is ca 80% male. Does this capture sex differences in NAC metabolism or toxicity patterns? This should be adressed in the discussion, as these differences occur throughout metabolism; there are documented effects on hormone metabolism and on circadian cycles, at least.
Author Response
Reviewer #3
This work by Meadows and colleagues adresses the use of NAC to reduce kidney and liver injury in MDR-TB patients, presenting a randomized trial investigating whether NAC mitigates adverse events or not.
The authors provide a well-structured background, clearly defining the rationale for NAC use. The methodology is generally well-detailed, and the study is adequately powered for exploratory analysis.
Reply: Thank you for recognizing the care in structuring the rationale of the study and the rigorous design to answer such a clinically relevant question.
Some issues need to be clarified at this stage, please see below.
Major comments
- How were AE identified? Specifically, how did the atuhors identify situations of "renal toxicity, liver toxicity, anemia"? Through blood works? Other? This needs to explicited, I could only find information regarding the classification of the severity of the AE.
Reply: Thank you for your thorough review. We agree with importance of clearly stating the criteria for AE and how the clinical and laboratory data for AEs were collected- the criteria are discussed in lines 193 to 197 of the manuscript. Additionally, we now cite Supplemental Table 1, which provides more detailed information. For data collection, we assessed clinical history (patient report and physical examination) and blood work (comprehensive metabolic panel and complete blood count) at baseline and all follow-up visits—these methods have been explicitly stated in our manuscript; please refer to lines 190-205 for further details.
- No statistically significant difference was found between the various groups, which in itself is OK, but the authors highlight a “trend” toward renal toxicity reduction with NAC (23% vs. 45%, p = 0.058). The confidence interval for this p value likely includes the possibility of both no effect and harm. The authors should revisist the statistic of this study, presenting for example confidence intervals, without which it is difficult to assess the precision of estimates.
Reply: Our study was powered to detect the primary outcome, which was to compare the total number of adverse events between the NAC and standard treatment groups. While the difference was not statistically significant, we observed a decrease in renal toxicity (referring to this as a “trend given p-value from 0.1-0.050- in this case 0.058) for this planned secondary analysis of total events of nephrotoxicity in those that received NAC and those that did not (and as the Reviewer points out a considerable reduction—23% of 44 patients in the NAC group compared to 45% of 22 patients in the standard treatment group—but a finding which may still be spurious). While given the directionality it is unlikely that NAC would be causing harm (more events of nephrotoxicity in the NAC groups if conducted over a larger study), we agree that this is exactly what a new randomized controlled trial could be designed to detect, that is powering on the primary outcome of nephrotoxicity based on the observed effect size in this first/current trial.
- Also, the study population is ca 80% male. Does this capture sex differences in NAC metabolism or toxicity patterns? This should be adressed in the discussion, as these differences occur throughout metabolism; there are documented effects on hormone metabolism and on circadian cycles, at least.
Reply: We agree with this potentially important effect. In lines 338–341 of the manuscript, we address the potential impact of gender on liver toxicity, referencing Zhao H et al. (2020), a retrospective study that supports this association. However, we recognize that the metabolism and toxicity of RR/MDR-TB treatment may vary by sex due to differences in body composition, enzyme activity, and hormonal regulation. Previous studies have reported that women tend to have higher plasma concentrations of certain anti-TB drugs, such as rifampicin and isoniazid, potentially increasing their risk of drug-induced toxicity. Additionally, sex-related differences in renal clearance and liver enzyme activity may influence drug metabolism and adverse event profiles. Since our study population was approximately 80% male, it may not fully capture these potential sex-based variations and we now make sure to state this a limitation (see Discussion, limitations paragraph). Future studies with a more balanced gender distribution would be needed to determine whether sex influences the efficacy and safety of RR/MDR-TB treatment and if NAC’s potential protective effect performs differently by sex.
Round 2
Reviewer 1 Report
Comments and Suggestions for Authors
My concerns have been addressed.
Reviewer 2 Report
Comments and Suggestions for Authors
The authors have satisfactorily addressed most of my concerns in a thorough and satisfactory fashion. I consider the manuscript acceptable for publication.
Comments on the Quality of English Language
Still some typos are there which could have been corrected before final submission.
Reviewer 3 Report
Comments and Suggestions for Authors
I would like to thank the authors for the edits and their responses.
I believe the manuscript can be published in its present form.